# Transformers to SSMs: Distilling Quadratic Knowledge to Subquadratic Models

**Aviv Bick**[1][*], **Kevin Y. Li**[1][*], **Eric P. Xing**[12], **J. Zico Kolter**[1], **Albert Gu**[13]
[1]Carnegie Mellon University, [2]MBZUAI, [3]Cartesia.ai
{abick, kyl2}@cs.cmu.edu

## Abstract

Transformer architectures have become a dominant paradigm for domains like language modeling but suffer in many inference settings due to their quadratic-time self-attention. Recently proposed subquadratic architectures, such as Mamba, have shown promise, but have been pretrained with substantially less computational resources than the strongest Transformer models. In this work, we present a method that is able to distill a pretrained Transformer architecture into alternative architectures such as state space models (SSMs). The key idea to our approach is that we can view both Transformers and SSMs as applying different forms of mixing matrices over the token sequences. We can thus progressively distill the Transformer architecture by matching different degrees of granularity in the SSM: first matching the mixing matrices themselves, then the hidden units at each block, and finally the end-to-end predictions. Our method, called MOHAWK, is able to distill a Mamba-2 variant based on the Phi-1.5 architecture (Phi-Mamba) using only 3B tokens. Despite using less than 1% of the training data typically used to train models from scratch, Phi-Mamba boasts substantially stronger performance compared to all past open-source non-Transformer models. MOHAWK allows models like SSMs to leverage computational resources invested in training Transformer-based architectures, highlighting a new avenue for building such models.

## 1 Introduction

Large language models based upon Transformer architectures have become a staple of natural language processing but suffer from their reliance on quadratic self-attention — the need to compute inner products between tokens at all positions up to the context length. This has motivated the development of several alternative subquadratic models, either approximations of self-attention [23] or entirely different architectures, such as state space models (SSMs) [13, 12, 30, 35]. Training strong subquadratic models such as SSMs can benefit the community through their cheaper finetuning and inference costs; however, they have not benefitted from the same amount of community effort in the form of training and compute as for Transformers. This raises a natural question: is it possible to leverage the vast amounts of resources that have been invested in training quadratic-time Transformers and use these models to produce stronger alternative models, such as state-space models?

In this paper, we present an approach for training subquadratic state-space models (specifically from the class of Mamba SSMs [12]) through the distillation of different elements of a pretrained Transformer model. The key intuition is viewing both Attention and SSMs as sequence transformations that mix different token embeddings by applying different classes of matrices across them. Sequence model *architectures* can then be factored into separate (i) sequence mixing and (ii) channel mixing blocks, e.g., a Transformer is composed of Attention (sequence mixer) and MLP (channel mixer)

---

[*]Authors contributed equally to this work.

38th Conference on Neural Information Processing Systems (NeurIPS 2024).

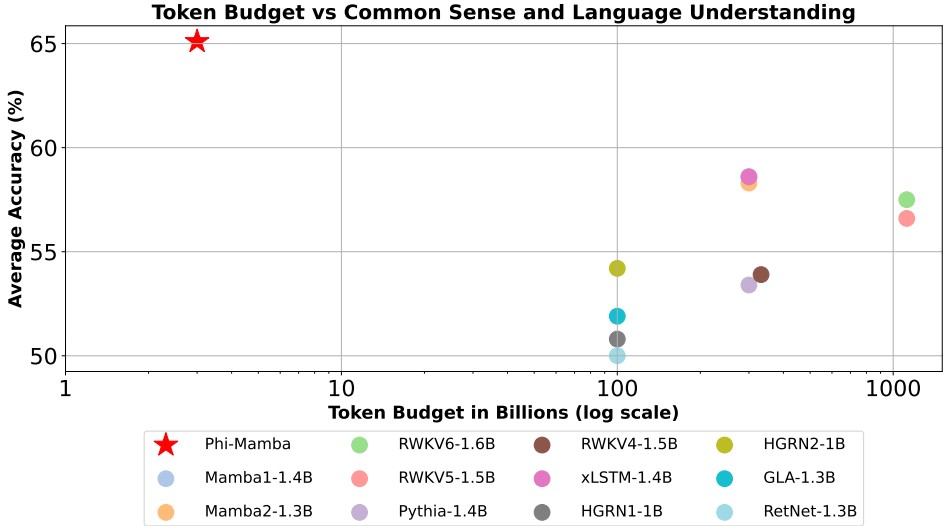

Figure 1: Plot of trained token budget to averaged accuracy on Winogrande, Arc-E, Arc-C, PIQA, and Hellaswag on various open-source models (mainly non-Transformer-based models). Our model (Phi-Mamba) uses more than $33\times$ less token budget to achieve 5% higher average accuracy than the next best model.

blocks. Using this breakdown, we can separately distill the *mixing* elements of each model explicitly at different levels of granularity. Specifically, we propose a three-phase distillation process that progressively targets higher levels of supervision from the teacher model: (1) a matrix orientation phase that aligns the sequence transformation matrices themselves; (2) a hidden-state distillation that aligns the hidden-state representations of each individual layer of the network without sacrificing pre-existing learned representations; and (3) an end-to-end training phase with weight transfer that finally distills the final output of the network using only a fraction of training data. We term our approach **MOHAWK** after these three stages (**M**atrix **O**rientation, **H**idden-State **A**lignment, **W**eight-Transfer and **K**nowledge Distillation).

We apply our approach to a modified instantiation of the Mamba-2 architecture [8], termed Phi-Mamba, which is aimed at more directly corresponding to the different architectural blocks of the Phi-1.5 language model [15] — a very strong Transformer model at the 1.3B parameter scale. Using our approach, the Phi-Mamba model achieves performance on benchmarks *stronger than any previous Mamba models of similar size*. Although performance still lags behind that of the base Phi-1.5 model on these benchmarks, the model is distilled with only 3.0B tokens, less than 1% of the data used to train either the previously best-performing Mamba models and 2% for the Phi-1.5 model itself. For instance, our Phi-Mamba achieves a 71.7% accuracy on the Winogrande dataset, compared to the pretrained Mamba-2 model's 60.9% accuracy, and 44.1% accuracy on the ARC-C dataset, compared to Mamba-2's 33.3% accuracy. Our results highlight the benefit of our three-phase distillation approach: we show in ablation experiments that each phase is highly beneficial for the eventual performance of the model, and that, e.g., *only* attempting to directly distill the Phi-1.5 model (i.e., Phase 3 alone) substantially underperforms the full MOHAWK method. Moreover, our findings emphasize the benefits of state-space models while training on fewer than $100\times$ tokens than the original pretrained Mamba model.

## 2 Related Work

**Sequence Models.** State-of-the-art autoregressive language models have been pretrained on massive amounts of data, resulting in models that exhibit extensive downstream capabilities, such as zero-shot translation and long-range reasoning [3, 15, 37]. Recent work has focused on addressing the quadratic complexity of Transformers by developing subquadratic alternatives based on RNN [30, 1], SSM [12, 35], and linear attention mechanisms [23, 43, 25, 7, 32], highlighting the importance of efficient sequence models in the era of large-scale autoregressive language models.

**Distillation.**    Knowledge distillation can be used to transfer knowledge from a large teacher model to a smaller student model, resulting in a more efficient model that retains the performance of the teacher model [18]. Distillation has been applied to various language modeling tasks, such as text generation [5, 17], machine translation [16, 48, 36], and question-answering system [19, 44].

Distillation in language models has been largely focused on *compression*: turning a larger pretrained Transformer into a smaller one by utilizing the weights of the teacher model [41, 21, 42]. Some of the techniques proposed look similar to ours; for example, [41] match attention matrices in a step similar to our matrix orientation, and [24] align outputs of each block (i.e., the hidden states). However, these differ in subtle and important ways because of our setting; for example, the former uses a different loss function than us that relies on softmax attention, and the latter is an end-to-end objective while our hidden state alignment occurs completely independently block-per-block. Consequently, prior work has observed that combining these objectives does not actually help or even hurts distillation [21], whereas we show that our techniques all significantly help improve the student model.

A smaller body of work has focused on our objective of distilling *across architectures*, in particular, turning a pretrained Transformer into a different architecture (usually some recurrent model) of the same size. [22] converted a pretrained softmax attention into linear attention by directly transferring weights and continuing fine-tuning. A similar approach was taken by concurrent works for converting Attention into linear RNNs [27, 40]. Recently, [47, 46] also proposed distilling into linear attention by first matching attention matrices. Our approach differs by using a different loss function that works beyond linear attention; incorporating more stages (e.g., the hidden state alignment step); and using recent, more expressive classes of efficient student models (Mamba-2), which we show are significantly easier to distill into (Table 4).

## 3    Preliminaries

To facilitate a clear understanding of our distillation approach, we start with the necessary background and definitions. An overview of the Mamba-2 architecture, which forms the foundation of our Phi-Mamba model, is also provided.

### 3.1    Matrix Mixers

Following [8], we refer to an equivalent function that represents the input and output of a sequence model as a *sequence transformation* or a *sequence mixer*. Formally,

**Definition 1** (Sequence Transformation). *We use the term to refer to a parameterized map on sequences $Y = f_\theta(X)$ where $\mathbf{X}, \mathbf{Y} \in \mathbb{R}^{(T,P)}$ and $\theta$ is an arbitrary collection of parameters. $T$ represents the sequence or time axis; subscripts index into the first dimension, e.g. $X_t, Y_t \in \mathbb{R}^P$.*

To put it differently, sequence mixers combine tokens at various time steps, facilitating the model's comprehension of temporal information and interactions. Sequence transformations form the foundation of deep sequence models, being integral components of neural network frameworks such as Transformers. A particular family of sequence transformations can be represented by $\mathbf{Y} = \mathbf{MX}$ for a matrix $\mathbf{M} \in \mathbb{R}^{(T,T)}$, which we refer to as a *sequence transformation matrix* or *matrix mixer*. An example of such a matrix mixer is the vanilla self-attention, Softmax($\mathbf{QK}^\top$), which is applied to the input-dependent $\mathbf{V}$ resulting in the familiar Softmax($\mathbf{QK}^\top$)$\mathbf{V}$. Similarly, Linear Attention [23] has a sequence transformation matrix of the form $\mathbf{K}^\top$. In addition, we can easily obtain their causal variants by multiplying by $\mathbf{L}$, a lower triangular matrix filled with 1s, to obtain $\mathbf{L} \circ$ Softmax($\mathbf{QK}^\top$) and $\mathbf{L} \circ \mathbf{QK}^\top$, respectively. Another example is a Toeplitz matrix $\mathbf{T}$ used to perform discrete convolution on input $\mathbf{X}$, resulting in $\mathbf{TX}$ [31].

A naive approach to computing the output of a sequence transformation is to multiply the input sequence $\mathbf{X}$ by the matrix $\mathbf{M}$. However, this approach has a time complexity of $O(T^2)$, which is prohibitive for long sequences. Subquadratic sequence transformations, such as Mamba-2, have been developed to address such inefficiencies through structured matrix multiplication.

### 3.2    Mamba-2

Mamba-2 [8], a type of structured state space models (SSMs) [13, 11], was recently introduced. Similarly to the original Mamba model [12], Mamba-2 uses a time-varying state-space model which

can selectively focus on or ignore inputs due to its input-dependent parameterization of the system components. The time-varying SSM is defined as follows:

$$h_{t+1} = \mathbf{A}_t h_t + \mathbf{B}_t x_t$$
$$y_t = \mathbf{C}_t h_t \tag{1}$$

Here, $\mathbf{B}_t$ and $\mathbf{C}_t$ are input-dependent projections of the system, as in Mamba-1; however, $\mathbf{A}_t$ is the identity matrix $\mathbf{I}$ multiplied by a scalar $\alpha_t$. The above formulation also differs from the previous one by treating the underlying sequence as originating from a discrete signal instead of a continuous one and therefore omits the sampling component $\Delta t$ from the original Mamba model.

Importantly, Mamba-2 draws a new connection between SSMs and Transformers, termed *Structured State Space Duality (SSD)*, which shows that a special case of SSMs can be viewed as a form of causal linear attention. In particular, fixing $\mathbf{A}_t = I$ (a further restriction of Mamba-2 to $\alpha_t = 1$) results in the formulation of causal linear attention [23] with the matrices $\mathbf{B}$ and $\mathbf{C}$ representing the projections of the key and the query, respectively, while the input projection $\mathbf{X}$ corresponds to the projection of the value.

**Mamba-2 as a matrix sequence transformation.** Inspired by the aforementioned connection between SSMs and Transformers, [8] shows that Mamba-2's *SSD* mixer family is equivalent to sequentially-semi-separable matrices [4]. Formally, the SSD mixer family can be represented as:

$$\begin{aligned} h_{t+1} &= \alpha_t \cdot I h_t + \mathbf{B} x_t \\ y_t &= \mathbf{C} \cdot h_t \end{aligned} \quad \Rightarrow \quad \begin{bmatrix} \alpha_1 & 0 & 0 & \cdots & 0 \\ \alpha_{2:1} & \alpha_2 & 0 & \cdots & 0 \\ \alpha_{3:1} & \alpha_{3:2} & \alpha_3 & \cdots & 0 \\ \vdots & \vdots & \vdots & \ddots & \vdots \\ \alpha_{n:1} & \alpha_{n:2} & \alpha_{n:3} & \cdots & \alpha_n \end{bmatrix} \circ (C \cdot B^\top) \cdot X \tag{2}$$

where $\alpha_{t:i} = \alpha_{t-1} \cdot \alpha_{t-2} \cdots \alpha_i$. An interesting observation is that the Mamba-2 architecture can be viewed as a causal linear attention with a *learnable causal mask*.

# 4 Methods

Throughout this section, we will describe each phase of MOHAWK. Specifically, we will cover the stages of matrix orientation, hidden-state alignment, and knowledge distillation, all three of which are crucial for developing an effective student model from the pretrained Transformer model. Unlike traditional distillation techniques, the student model retains the overall architecture of the teacher model, differing only in the replacement of the attention matrix mixer with a subquadratic alternative. We will progressively unveil our architecture, Phi-Mamba, along with the specifics of its distillation process. This section concludes with an in-depth description of the Phi-Mamba architecture and its hybrid version, which surpasses the performance of other subquadratic matrix mixers. Further examinations of the effectiveness of the method and ablation studies are discussed in Section 5.

For clarity, the term *block* refers to the repeating components that form the end-to-end model. The blocks are composed of *layers*, such as the self-attention layer (including projections), the SSM layer (including the mixer and convolution), and the convolutional layer. In this manner, many Transformer models, like Llama [37], are viewed as a stack of alternating self-attention and MLP blocks, whereas the Phi and Phi-Mamba models are comprised of Phi blocks that have parallel Attention/SSM and MLP blocks.

## 4.1 Stage 1: Matrix Orientation

The first stage of MOHAWK aims to align the student matrix mixer with the teacher's self-attention matrix. Achieving this alignment is a two-step process: first, at every mixing layer, the student components preceding the matrix mixer are set to match the teacher's components. This ensures that each layer's input undergoes the same transformation up to the matrix mixer section. Consequently, the only variation from the input to the mixing process is the matrix calculation. We then minimize the distance between the matrix mixer, e.g., the self-attention matrix and the materialized SSM matrix (2), of each layer within the student and teacher models:

$$\min_{\phi} \|\text{TeacherMixer}(\mathbf{u}) - \text{StudentMixer}_{\phi}(\mathbf{u})\|_F \tag{3}$$

where $\phi$ denotes the parameters within the student's sequence mixing layer, and $\mathbf{u}$ indicates any arbitrary input. In our experimental setup, $\mathbf{u}$ was chosen as the output from the teacher model's preceding layer to better mimic the input distribution to the layer. This stage ensures that the student and teacher models have roughly similar mixing layers and sets the foundation for the subsequent stages of the distillation process. In particular, this stage can be done in parallel across all the student layers, as the inputs to the student and teacher blocks are identical.

For Mamba-2, we begin by setting the convolution to an identity function, effectively nullifying its initial impact. This results in the computation of the semi-separable matrix being the sole distinction between the layers. We then proceed to minimize the distance between the two matrix mixers: the semiseparable scalar identity and the attention matrix (see Figure 2). Figure 3 demonstrates the importance of this stage in the distillation process. Furthermore, Table 3 shows that the Mamba-2 matrix mixer is more expressive than popular alternatives and can closely approximate the self-attention matrix of various data samples across all layers of a Transformer model through gradient descent, solidifying it as a strong sequence mixer.

### 4.2 Stage 2: Hidden-State Alignment

Following the optimization of Equation (3), we must still address the differences between the outputs of the student and teacher blocks. To achieve this, we further align the components of the two blocks using initialization and distillation. Specifically, our goal is to match each student and teacher mixing blocks by minimizing the L2 norm of their output (e.g., the entire Mamba block with the self-attention block):

$$\min_{\phi} \|\text{AttnBlock}(\mathbf{u}) - \text{StudentMixerBlock}_{\phi}(\mathbf{u})\|_2 \tag{4}$$

where similar to Section 4.1, $\phi$ represents student's block parameters, and $\mathbf{u}$ is an input. Once again, this stage can be done in parallel across all the student layers.

In the case of Mamba-2, we modify the remaining components to be identical to the Phi-1.5's Attention block, so that the overall functionality is preserved from Stage 1. Concretely, we initialize the gate (see Figure 2) to a constant value of 1 to "open" the gate, canceling its initial effect. In addition, we remove the normalization prior to the output projection, as it cannot be set to align with the Attention block. We then minimize the distance between the output of the Mamba-2 block and the output of the teacher's self-attention block. Our analysis indicates that the distance between the Mamba-2 block and the self-attention block is strongly correlated with the model's ability to learn the teacher's distribution, as shown in Table 3. Furthermore, Figure 3 shows that a better independent alignment of the student and teacher blocks results in performance improvements, highlighting the importance of this stage in the distillation process.

### 4.3 Stage 3: Weight-Transfer and Knowledge Distillation

The final stage of the distillation process aims to fine-tune the student model to match the performance of the teacher model. Although each student mixing block is aligned with its corresponding teacher mixing block, discrepancies are still present between consecutive blocks throughout the network To bridge these gaps and address the remaining components of the language model, we transfer the remaining weights of the teacher model to the student's respective components. For Phi-Mamba, this involves the token embedding, the final layer normalization, the Language Model head, and the MLP and input norm at each block (see Figure 2). We then fine-tune the complete end-to-end student model under teacher supervision. Concretely, we use a distillation loss to encourage the student model to mimic the distribution of the teacher model's logits, also known as knowledge distillation [18]:

$$\min_{\phi} \mathcal{L}_{\text{CE}}\big(\text{TeacherModel}(\mathbf{x}), \text{StudentModel}_{\phi}(\mathbf{x})\big) \tag{5}$$

where $\mathbf{x}$ is the input tokens to the models.

It has been hypothesized that much of the information stored in language models resides in MLP blocks [28]. To utilize the work already done pretraining the teacher, MOHAWK adjusts the structure of the student blocks to utilize the MLP in the same way as the teacher model, effectively swapping the teacher's matrix mixer with that of the student.

Interestingly, during this step, the MLP weights *can be kept frozen* while keeping the model performant. This showcases Mamba-2's powerful expressiveness crucial for replacing Attention, cuts the

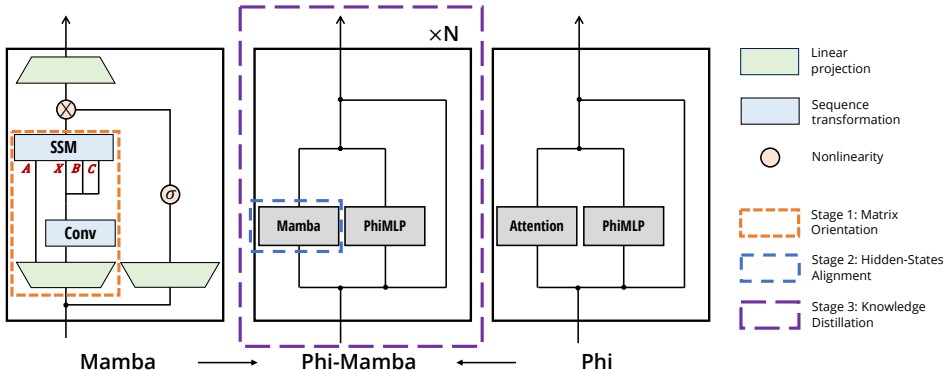

Figure 2: The Phi-Mamba architecture consists of a stack of blocks, each of which contains a Mamba block and an MLP block. The Mamba block is a simplified version of the Mamba-2 block [8] that omits the non-linear activation function after the convolutional operation and the layer normalization present before the output projection, so that the parts of the model outside the matrix mixer can be transferred from the teacher model. The MOHAWK distillation process involves progressively matching fine-to-coarse parts of the model to the corresponding part of the teacher model: (1) the mixer mixer itself (2) the full Mamba vs. Attention blocks, and (3) the end-to-end model.

number of trained parameters by more than half, and, in larger models, helps prevent the student model from experiencing catastrophic forgetting of the teacher model's information. We validate Mamba-2's ability to do so in Table 5.

## 4.4 Phi-Mamba architecture

Combining the three stages of MOHAWK, we introduce the *Phi-Mamba* architecture, which merges the Mamba-2 model of [8] with the Phi-1.5 Transformer model of [15]. It consists of a stack of Phi-Mamba blocks (Figure 2), initialized and distilled as described in previous sections.

Overall, the Phi-Mamba architecture, as depicted in Figure 2, differs from the vanilla Mamba-2 architecture by modifying the structure of the SSM matrix mixer, removing components from the SSM block and incorporating dense layers from the teacher model. In particular, each Mamba-2 block was modified by removing post-convolution activation and pre-output projection normalization, while setting the gate and convolution to be identity functions. Interestingly, although these components were found to be beneficial for performance when Mamba-2 was trained from scratch [8], we find that they are unnecessary for our distillation process.

Two key changes were made to the Mamba-2 matrix mixer. The first was converting the SSM head structure from multi-value to multi-head, much like the multi-head attention mechanism found in Transformers [39], enabling the independent distillation of each Transformer head into a Mamba head. Moreover, we handle the sequence mixer as entirely discrete-time by making the $\mathbf{A}$ matrix a projection of the input and eliminating the $\Delta$ discretization parameter. Although this formulation slightly differs from Mamba-2, the original algorithm can still be applied as a black-box method.

## 5 Empirical Validation

We empirically validate the MOHAWK framework is able to achieve better performance on various downstream benchmarks compared to previous subquadratic models of similar size. Our final Phi-Mamba-1.5B model is distilled on 3 billion tokens (distributed as 80M in Stage 1, 160M in Stage 2, and 2.76B tokens in Stage 3 as described in Appendix A) from the C4 dataset, with a sequence length of 2048. This constitutes less than 1% of the resources used by many top-performing subquadratic open-source models (e.g., the original Mamba-1/2 models pretrain on 315B tokens).

Table 1 presents a comprehensive breakdown of downstream evaluation results for our models and multiple baselines on a standard set of commonsense reasoning and language understanding tasks: WinoGrande [33], HellaSwag [45], PIQA [2], ARC-Challenge and ARC-Easy [6], and

Table 1: Downstream evaluation results for full methods, comparing Phi-Mamba against open-source models of similar sizes pretrained on standard language modeling corpuses. Phi-Mamba attains performance close to the teacher model and better than all pretrained models, while using less than 1% of the training data.

| MODEL | TOKENS | WINOG. | ARC-E | ARC-C | PIQA | HELLAS. | LAMB. | AVG. ↑ |
|---|---|---|---|---|---|---|---|---|
| Phi-1.5-1.3B | 150B | 73.4 | 75.6 | 48.0 | 76.6 | 62.6 | 53.4 | 64.9 |
| **Phi-Mamba-1.5B** | 3.0B | **71.7** | **74.0** | **44.1** | **75.5** | 60.2 | 50.1 | **62.6** |
| Mamba-1-1.4B | 315B | 61.5 | 65.5 | 32.8 | 74.2 | 59.1 | 64.9 | 59.7 |
| Mamba-2-1.3B | 315B | 60.9 | 64.3 | 33.3 | 73.2 | 59.9 | 65.7 | 59.6 |
| Finch-1.6B | 1.1T | 59.4 | 64.2 | 34.1 | 72.6 | 57.3 | **66.8** | 59.1 |
| xLSTM-1.4B | 300B | 60.6 | 64.3 | 32.6 | 74.6 | **60.9** | 57.8 | 58.5 |
| Eagle-1.5B | 1.1T | 59.1 | 64.3 | 33.5 | 71.1 | 55.0 | 65.7 | 58.1 |
| Pythia-1.4B | 300B | 57.3 | 60.6 | 26.0 | 71.1 | 52.1 | 61.6 | 54.8 |
| RWKV4-1.5B | 330B | 54.6 | 60.5 | 29.4 | 72.4 | 52.5 | 56.4 | 54.3 |
| DeltaNet-1.3B | 100B | 53.6 | 57.2 | 28.3 | 71.2 | 50.2 | 48.9 | 51.6 |
| GLA-1.3B | 100B | 53.9 | 57.2 | 26.6 | 71.8 | 49.8 | 46.9 | 51.0 |

Table 2: MOHAWK distillation from Phi-1.5 teacher model to Phi-Mamba-1.5B. "Stages Applied" details which of the three MOHAWK stages was performed, highlighting the importance of each stage. All experiments executed using a fixed amount of 5B tokens for the entire distillation process.

| STAGES TYPE | WINOG. APPLIED | ARC-E ACC ↑ | ARC-C ACC ↑ | PIQA ACC ↑ | HELLAS. ACC ↑ | LAMB. ACC ↑ | AVG. ACC ↑ |
|---|---|---|---|---|---|---|---|
| 2 | 55.9 | 75.4 | 38.0 | 75.2 | 56.6 | 18.9 | 53.3 |
| 3 | 62.8 | 64.3 | 27.8 | 75.6 | 52.6 | 43.8 | 54.5 |
| 2-3 | 72.3 | 75.0 | 40.8 | 75.2 | 59.7 | 50.6 | 62.3 |
| 1-3 | 74.8 | 72.7 | 43.5 | 75.6 | 59.6 | 49.2 | 62.7 |

LAMBADA [29]. Figure 1 shows the performance versus the training cost of Phi-Mamba compared to many open-source baselines from the literature at similar model sizes.

For the remainder of this section, we will analyze the impact of the 3 stages of MOHAWK one by one. Throughout the experiments detailed in this section, we use the AdamW optimizer with $\beta = (0.9, 0.95)$, a weight decay of 0.1, and a learning rate of $1 \times 10^{-4}$, combined with a Warmup-Stable-Decay (WSD) scheduler featuring 10% warmup and 10% decay. The training law figures and the final Phi-Mamba model use the regime detailed in Appendix A.

## 5.1 Stage 3 (Weight-Transfer and Knowledge Distillation)

As described in Section 4.3, this phase employs a simple end-to-end distillation of teacher-model logits. It leverages the alignment among all sequence mixers and successive blocks to jointly fine-tune all components of the network. Experiments shown in Table 2 highlight the relevance of implementing this end-to-end alignment, with all three architectures achieving their highest scores only after this phase. Predictably, the impact of end-to-end alignment varies by architecture: models with more mixing layers similar to the teacher model see a reduced importance of this phase.

Stage 3 is the only stage in MOHAWK that trains the student model end-to-end and can be seen as the "main" stage. Many distillation methods employ only this stage; however, Table 2 shows that using only end-to-end knowledge distillation is less than ideal. Although it is slightly advantageous to use only Stage 3 compared to only Stage 2, there is a significant gap between using only Stage 2 versus using Stage 2 + 3. As elaborated in Section 6, this phase can freeze all network components except the Mamba-2 sequence mixer without a significant performance drop. This in particular indicates that the third stage (like the other stages of MOHAWK) can operate in computationally limited settings.

## 5.2 Stage 2 (Hidden-State Alignment)

Following the analysis of the model's end-to-end distillation in Stage 3, we evaluate the impact of aligning the hidden-state outputs of mixer blocks (Stage 2) on both the subsequent Stage 3 process

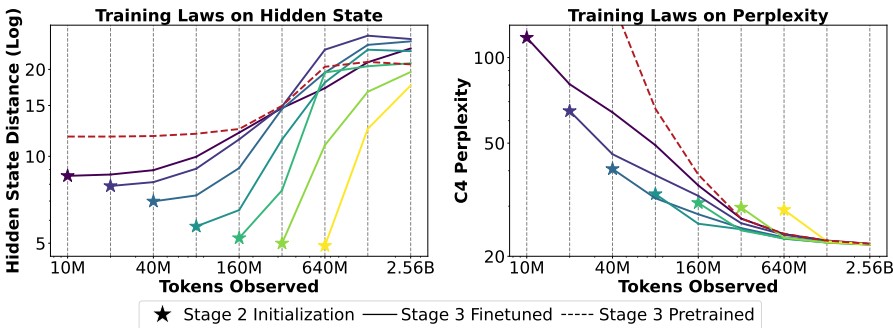

Figure 3: Training laws comparing the token budget between Stages 2 and 3, as measured by the Stage 2 metric (hidden state distance) and Stage 3 metric (perplexity). Stage 2 initializations are used as the starting checkpoint for their respective Stage 3 finetuning models. Stage 3 pretrained is trained from scratch only with weight transfer and knowledge distillation. Despite training for less tokens on Stage 3 than the Stage 3 from scratch, almost all Stage 2 initialized models eventually outperform the baseline in perplexity on a fixed budget. In general, better aligned Stage 2 initializations improve post-Stage 3 performance.

and overall downstream model performance. We accomplish this by training Phi-Mamba instances from scratch using Stage 2 to various token counts. From these checkpoints, we proceed to Stage 3 training, ending with different total budgets to allow us to analyze how the degree of Stage 2 "pretraining" impacts Stage 3 performance at various token budgets.

Figure 3 demonstrates that given an adequate training budget, models beginning with weights with lower hidden state distances (after Stage 2) outperform those that depend exclusively on knowledge distillation (Stage 3). These lower hidden states are also correlated with lower starting perplexities, which in turn are correlated with downstream performance, as shown in Figure 5. Furthermore, Table 2 shows the synergy between Stage 2 and Stage 3, as applying Stage 3 on top of Stage 2 outperforms vanilla knowledge distillation, highlighting the importance of incorporating both hidden-state alignment and knowledge distillation methods for the tested architectures.

### 5.3 Stage 1 (Matrix Mixer Orientation)

Motivated by our previous finding, we then analyze how matching the matrix mixers can decrease the overall mixer block's hidden-state distance with the teacher model even further. Similarly to our previous protocol, we assess the positive impact of the current stage on the following phase's metrics and final model's performance by comparing models with varying amount of Stage 1 and Stage 2 training on both stage metrics.

Figure 4 shows that even with constrained budgets, performing Stage 1 for a small period can help with subsequent stages and their performances. Thus, even a small amount of Stage 1 training can help their respective Stage 2 models reach better hidden-state distances compared to the from-scratch counterpart. This is despite the phenomenon that the teacher and student mixers diverge and then re-converge in Stage 2 after mixer similarity is no longer directly optimized. Coupled with Section 5.2, which discovers that lower hidden state initializations lead to better perplexity and downstream performance, it can be inferred that Stage 1 aids the overall distillation process. We further empirically validate this intuition in Table 2, which indicates that this stage aligns the matrix mixers to a stronger degree than only the hidden state alignment.

## 6 Self-Attention Approximation with Structured Matrix Mixers

We start by testing the ability of various matrix mixer families to match the empirical self-attention matrices of a pretrained Transformer. We take 1000 samples from each layer of a Llama2-7b-Chat model [37], materialize the attention matrices, and project them onto given classes of structured matrices. The results in Table 3 are averaged across all layers.

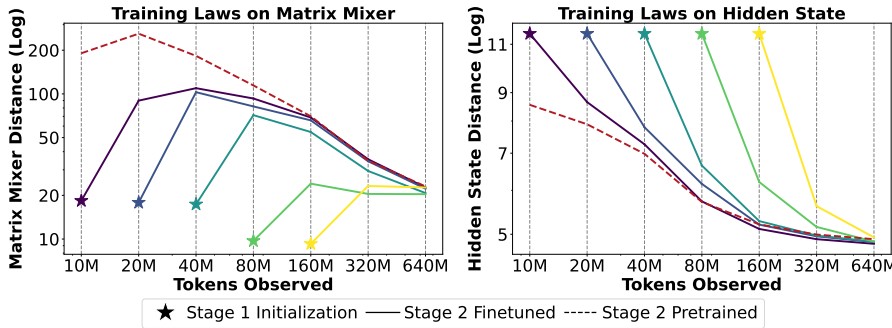

Figure 4: Training laws comparing the token budget between Stages 1 and 2, as measured by the Stage 1 metric (matrix mixer distance) and Stage 2 metric (hidden state distance). Even a small amount of Stage 1 training can improve the model's hidden-state distances in subsequent stages. Notably, this improvement occurs despite an increase in matrix mixer distance during Stage 2. This suggests that early Stage 1 training provides a foundational benefit that enhances the model's performance in later stages, demonstrating the importance of initial training phases in model optimization.

Table 3: Attention matrix approximation by structured matrix mixers (Frobenius distance; lower is better). Structures are Toeplitz, low-rank (LR), state space dual (SSD) model (3.2) and general semi-separable matrices (SSM), all causal. We used 1,000 samples, each 512 tokens. Samples were passed through Llama2-7B-Chat, and one attention head from each layer was randomly chosen for approximation. We evaluated (LR) and SSD families with 10,000 gradient descent steps per sample.

| STRUCTURE (*State size N*) | TOEP. - | LR (16) | SSD (16) | SSM (16) | LR (32) | SSD (32) | SSM (32) | LR (64) | SSD (64) | SSM (64) |
|---|---|---|---|---|---|---|---|---|---|---|
| WT-103 | 12.0 | 0.619 | 0.477 | 0.266 | 0.322 | 0.237 | 0.127 | 0.132 | 0.097 | 0.046 |
| OWT | 12.2 | 0.606 | 0.466 | 0.259 | 0.314 | 0.231 | 0.123 | 0.129 | 0.095 | 0.045 |
| C4 | 12.3 | 0.595 | 0.453 | 0.236 | 0.310 | 0.226 | 0.112 | 0.128 | 0.093 | 0.041 |
| IMdB | 12.3 | 0.598 | 0.455 | 0.238 | 0.312 | 0.226 | 0.113 | 0.129 | 0.094 | 0.043 |

In particular, to describe the class of linear attention matrices (3.1), we use the fact that $\mathbf{Q}$ and $\mathbf{K}$ are projections of the input $x \in \mathbb{R}^{d_{in}}$ onto $\mathbb{R}^{d_{out}}$, and therefore their rank is bounded by $\min \{d_{in}, d_{out}\}$. For multihead linear attention, $d_{out}$ (also known as head dimension) is typically a small value (e.g., Phi-1.5 and Llama2-7b-Chat have head dimensions of $64$ and $128$, respectively). Thus, we approximate this family of sequence mixers using causal low-rank matrices $\mathbf{L} \circ \mathbf{Q}\mathbf{K}^{\top}$, where $\mathbf{L}$ is a lower-triangular causal mask of 1s, and $\mathbf{Q}$, $\mathbf{K}$ are in $\mathbb{R}^{n \times d}$ with $d \ll n$ (indicating that the head dimension is substantially smaller than the sequence length).

To describe the multi-head Mamba-2 matrix family, we utilize the state space dual (SSD) layer (3.2) in a manner similar to the previous linear attention, but now the causal matrix $\mathbf{L}$ possesses an $n$-degree rolling multiplicative structure for SSD which can be seen as a more expressive mask that generalizes the causal mask (Section 3.2).

Both causal low-rank and SSD matrix families were approximated with 10,000 steps of gradient descent per sample. To approximate the general class of SSM matrix mixers, we utilize *balanced truncation*, a gradient-free projection algorithm. This method is mainly known in the field of time-invariant Dynamical System model reduction [14] and has been modified for use in time-varying systems [34]. Similarly, for the family of causal TOEPLITZ matrices, representing convolution operations, we employ a simple heuristic that minimizes the error for each attention matrix.

Table 3 shows that while the SSM matrix family provides the closest approximation to the self-attention matrix mixer, the Mamba-2 mixer family (SSD) has just twice the distance from the SSM matrices. This is in contrast to Linear Attention, which has three times the distance, all while keeping a computational cost on par with Linear Attention. More details can be found in Appendix C.

We further validate the ability of a Mamba-2 block to replace an Attention layer within a language model. Firstly, we create two variants of our architecture, Phi-Toeplitz and Phi-LR, and run the MOHAWK process for 1B tokens at each stage (see Table 4) to verify that the previous finding

Table 4: Ablations of matrix structure using the same training recipe (Stages 2 and 3). While many efficient sequence models (e.g. global convolutions, linear attention, and state space models) can be represented as structured matrix mixers (e.g. Toeplitz, low-rank, and semi-separable matrices respectively), more expressive structured matrix families can match the attention matrix more closely.

| MATRIX STRUCTURE | BLOCK OUTPUT L2 DIST. ↓ | WINOG. ACC ↑ | ARC-E ACC ↑ | ARC-C ACC ↑ | PIQA ACC ↑ | HELLAS. ACC ↑ | AVG. ACC ↑ |
|---|---|---|---|---|---|---|---|
| Causal Toeplitz | 9.6 | 49.3 | 21.2 | 26.2 | 52.3 | 25.9 | 35.0 |
| Causal low-rank | 7.6 | 50.2 | 27.9 | 25.0 | 53.3 | 25.6 | 36.4 |
| SSD | **5.5** | **67.2** | **71.0** | **38.6** | **74.2** | **45.0** | **59.2** |

Table 5: MOHAWK distillation for Phi-Mamba-1.5B on the entire model vs just the Mamba-2 component, i.e., frozen MLP, embedding, etc. MOHAWK can be used on just the sequence mixer blocks while keeping all other components frozen without compromising performance (Section 5.1).

| TRAINABLE COMPONENTS | WINOG. ACC ↑ | ARC-E ACC ↑ | ARC-C ACC ↑ | PIQA ACC ↑ | HELLAS. ACC ↑ | LAMB. ACC ↑ | AVG. ACC ↑ |
|---|---|---|---|---|---|---|---|
| All | 74.8 | 72.7 | 43.5 | 75.6 | 59.6 | 49.2 | **62.7** |
| Mamba-2 | 69.1 | 73.5 | 43.8 | 74.7 | 59.3 | 48.2 | **61.4** |

hold in a multilayer, end-to-end model case. Secondly, we run MOHAWK while freezing various parts of the Phi-Mamba modules (refer to Table 5), revealing that limiting the trainable elements to the Mamba-2 blocks (excluding the embedding, head and all MLP layers) results in only a minor performance decrease during MOHAWK distillation.

Interestingly, in all of the aforementioned experiments, we have found a consistent correlation between the projection distances of the matrix (Frobenius distance) in Table 3 and the downstream performance metrics (accuracy) in Table 4. Essentially, a better matrix approximation (lower Frobenius distance) is correlated with better model performance (higher accuracy) on various tasks. This connection highlights the relationship between the quality of the matrix approximation and the performance of the model. Such findings are echoed in [20], which find that more expressive matrix mixers lead to more performant models, e.g., Low-rank-based BERT models outperform Toeplitz-based ones.

# 7    Discussion and Conclusion

Our experiments shows that the Mamba-2 model can be successfully distilled from a pretrained Transformer teacher model, utilizing its extensive knowledge learned from custom datasets and higher computational resources. Despite using less than $100\times$ data compared to many open-source models, including Mamba, our subquadratic model outperforms other subquadratic models in various benchmark tests by a wide margin.

The MOHAWK framework's multi-stage process which gradually increased the scope of distillation is essential extracting the teacher model's knowledge to the fullest extent as shown in our ablations and training laws. We continue to find the effectiveness of MOHAWK when distilling hybrid Attention-SSM models and provide ablations on the number and position of Attention layers.

Additionally, we demonstrate that Mamba-2's relationship to Transformers is evident not only in theory, but also in practice, as it captures interactions similar to those of Transformers, and is able to replace Attention with little drop in performance. Coupled with past research which has posited that much of a language model's knowledge is embedded in the MLP blocks, we believe that any subquadratic model with a sufficiently expressive matrix mixer can replicate the behavior of pretrained Transformers, bringing quadratic knowledge to subquadratic models. We recommend further research to explore the role of sequence mixing layers in subquadratic models and their impact on performance. Advancements in both the distillation process and the sequence mixer architecture could lead to further improved performance in a range of tasks. We propose that "trainability" and "distillability" are distinct properties of the models, and therefore, distillation techniques should be more appropriately tailored to the model.

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

# A  Experiments and Experimental Details

**Hyperparameter Search**   To construct Appendix A, we performed grid searches for training in Stages 1, 2, and 3 independently from scratch to find the optimal hyperparameters. We explored learning rates $\text{lr} = \{1, 2, 5\} \times 10^{\{-3, -4\}}$ and batch sizes $2^{\{15, 16, 17, 18\}}$. AdamW Optimizer was used with $\beta = (0.9, 0.95)$, incorporating a weight decay of 0.1, gradient clipping at 1.0, and a Warmup-Stable-Decay (WSD) scheduler with 10% warmup and 10% decay utilizing linear warmup and cooldown functions. Automatic mixed precision training to bf16 was used in all stages. For Stages 1 and 2, we initially fixed the batch size at $2^{16}$, then varied the learning rates. After identifying the optimal learning rate, we adjusted the batch sizes and subsequently finalized the learning rate after fixing the batch size. Consequently, Stage 1 used $\text{bs} = 2^{15}, \text{lr} = 5 \times 10^{-4}$ and Stage 2 used $\text{bs} = 2^{15}, \text{lr} = 2 \times 10^{-3}$. In Stage 3, we set the batch size to $2^{19} \approx 0.5\text{M}$ and focused solely on varying the learning rate, resulting in $5 \times 10^{-4}$. Stages 1 and 2 were trained to 200M steps each while Stage 3 extended to 1B steps. For the Phi-Mamba ultimate model, the Stage 3 learning rate was reduced to $2 \times 10^{-4}$ to enhance stability.

**Multi-Stage Distillation Procedure**   In the development of the training law (see Figure 3), we executed a single "continuous" run initialized from a state that included several checkpoints. The warm-up period was determined as 10% of the tokens processed during the continuous run. For instance, if the model's goal was to process 640 million tokens, and it started from a run that had processed 40 million tokens, then the warm-up would be set at 60 million tokens. The checkpoints recorded during the warm-up phase were preserved as they were, while subsequent checkpoints underwent a cooling of 10% of the current phase. To illustrate, in the scenario mentioned earlier, a checkpoint at 320 million tokens during the 40M to 640M run would maintain the original warmup, while the cooldown would span 28 million tokens. Conversely, a checkpoint at 80 million tokens within the warm-up phase would be saved without any cooldown.

**Training Laws on Downstream Metrics**   Figure 5 extends the Stage 2 versus Stage 3 comparison in Figure 3, except we measure average accuracy on downstream metrics instead of perplexity. We observe a strong correlation between the training laws of perplexity and downstream evaluation metrics. While the general trend indicates that models exposed to more tokens during the prior stage initialization tend to perform better on both perplexity and downstream metrics, the relationship is not perfectly aligned. Specifically, the order of model performance based on perplexity does not always match the order based on downstream metrics, highlighting some differences in how these metrics capture model effectiveness.

**Training the Final Phi-Mamba Model**   After confirming the importance of the stages in Section 5.1, Section 5.2, and Section 5.3, we proceed to distill the final Phi-Mamba model using the three elements of MOHAWK. We use 80M tokens for Stage 1, due to the strong performance of the token count in both the matrix and hidden state distances (Figure 4). Stage 2 was distilled for 160M tokens given the apparent saturation of both hidden state distance and perplexity compared to the other initialization states, such as 10M, 20M, 40M, etc. (Figure 3). We employed Stage 3 to a total of 3B tokens across all stages and observed that the previously optimal learning rate applied for training training laws led to instabilities in training, particularly spikes in evaluation perplexity. Decreasing the learning rate for Stage 3 mitigated this issue as mentioned above. We hypothesize that the instability is due to the Stage 1 + 2 initialization's Mamba component being quite similar to that of the teacher model, so a large learning rate coupled with disconnect between blocks, which are mended in Stage 3, can cause training instabilities. The performance of the final model is reported in Table 1.

# B  Applying Mamba-2 as a Black Box

As noted previously Section 4.4, our Mamba-based sequence mixer is slightly modified from the original to make it more amenable for distilling from a Transformer architecture. In particular, the Mamba-2 sequence mixer is treated entirely in discrete time by projecting the input onto the matrix $\mathbf{A}$ and removing the discretization parameter $\Delta$. Even though this formulation is somewhat different from Mamba-2, the original algorithm remains applicable through a reduction expressed in Appendix B.

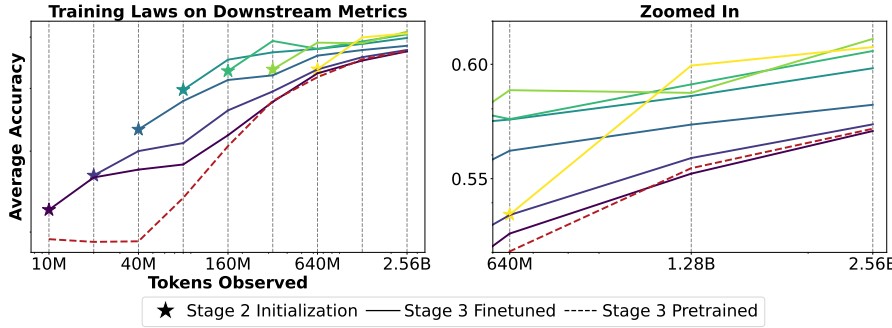

Figure 5: Training laws comparing the amount of token budget between Stages 2 and 3, as measured by the average accuracy of downstream evaluation metrics.

---

**Listing 1** PyTorch example for using the Mamba algorithm for a $Delta$-free variation.

```
"""
X: (batch, seqlen, nheads, headdim)
A_log: (batch, seqlen, nheads)
B: (batch, seqlen, nheads, dstate)
C: (batch, seqlen, nheads, dstate)
D: (nheads)
"""

y = Mamba(
    X = X / A_log.unsqueeze(-1),
    dt = rearrange(A_log, "b c h -> b h c"),
    A = torch.ones(self.nheads),
    B = B,
    C = C,
)
Du = torch.einsum("h,blhp->blhp", D, X)
y = rearrange(y + Du, "b l h p -> b l (h p)")
```

---

## C   Attention Matrix Approximation Details

This section serves as a complement to Section 6 and outlines the methods employed to create Table 3. Appendices C.1 to C.5 describe our strategies for finding a matrix within the specified families that closely approximates the original attention matrix using a selected distance metric. Formally, we consider the following optimization problem:

$$\min_{\mathbf{X}\in\mathcal{M}}\|\mathbf{M}-\mathbf{X}\| \tag{6}$$

where $\mathcal{M}$ is the subspace of a specific matrix family, $\mathbf{M}$ is the attention matrix, and $\|\cdot\|$ corresponds to a selected distance metric. In the following sections, we explore different methods and matrix families for this optimization problem.

### C.1   Semi-Separable Matrix Approximation

Considering a time-varying system denoted by $\{\mathbf{A_k},\mathbf{B_k},\mathbf{C_k},\mathbf{D_k}\}_{k\in[l]}$, we can describe it using the matrix mixer $T$ (also known as the transfer matrix) as follows:

$$T = \begin{bmatrix} D_1 & 0 & 0 & 0 & 0 & \cdots & 0 \\ C_2B_1 & D_2 & 0 & 0 & 0 & \cdots & 0 \\ C_3A_2B_1 & C_3B_2 & D_3 & 0 & 0 & \cdots & 0 \\ C_4A_{3:2}B_1 & C_4A_3B_2 & C_4B_3 & D_4 & 0 & \cdots & 0 \\ \vdots & \vdots & \vdots & \vdots & \vdots & \ddots & \vdots \\ C_lA_{l-1:2}B_1 & C_lA_{l-1:3}B_2 & C_lA_{l-1:4}B_3 & C_lA_{l-1:5}B_4 & \cdots & C_lB_{l-1} & D_l \end{bmatrix}$$

With $\mathbf{A_k}\in\mathbb{R}^{n\times n}$, $\mathbf{B_k}\in\mathbb{R}^{m\times n}$, $\mathbf{C_k}\in\mathbb{R}^{p\times n}$, and $\mathbf{D_k}\in\mathbb{R}^{p\times m}$, where $n$ is the state dimension, $m$ the input dimension, and $p$ the output dimension.

Table 6: Full attention matrix approximation by structured matrix mixers Structures are Toeplitz, causal low-rank (LR), RetNet, state space dual (SSD) model (3.2) with and without the diagonal **D** term and general semi-separable matrices (SSM). We have used 1,000 samples, each consisting of 512 tokens. Llama2-7B-Chat was applied on every sample, and one attention head from each layer was randomly chosen for approximation. We evaluated (LR), RetNet, and SSD families with 10,000 gradient descent steps per sample.

| STRUCTURE (State size $N$) | TOEPLITZ - | CAUSAL LOW-RANK (16) | RETNET (16) | SSD WITHOUT **D** (16) | SSD (16) | SEMI. SEP. MATRIX (16) |
|---|---|---|---|---|---|---|
| WT-103 | 12.0 | 0.619 | 0.530 | 0.522 | 0.477 | 0.266 |
| OWT | 12.2 | 0.606 | 0.516 | 0.508 | 0.466 | 0.259 |
| C4 | 12.3 | 0.595 | 0.503 | 0.496 | 0.453 | 0.236 |
| IMdB | 12.3 | 0.598 | 0.505 | 0.498 | 0.455 | 0.238 |
| (State size $N$) | - | (32) | (32) | (32) | (32) | (32) |
| WT-103 | 12.0 | 0.322 | 0.268 | 0.262 | 0.237 | 0.127 |
| OWT | 12.2 | 0.314 | 0.261 | 0.255 | 0.231 | 0.123 |
| C4 | 12.3 | 0.310 | 0.255 | 0.249 | 0.226 | 0.112 |
| IMdB | 12.3 | 0.312 | 0.256 | 0.251 | 0.226 | 0.113 |
| (State size $N$) | - | (64) | (64) | (64) | (64) | (64) |
| WT-103 | 12.0 | 0.132 | 0.110 | 0.107 | 0.097 | 0.046 |
| OWT | 12.2 | 0.129 | 0.107 | 0.104 | 0.095 | 0.045 |
| C4 | 12.3 | 0.128 | 0.106 | 0.102 | 0.093 | 0.041 |
| IMdB | 12.3 | 0.129 | 0.106 | 0.103 | 0.094 | 0.043 |

As **T**'s form corresponds to a semi-separable matrix (i.e., each sub-matrix has a rank of up to $n$), we will label this matrix form as *SSM with state size* $n$ throughout the remainder of this appendix, representing a state-space model with state size $n$ or a semi-separable matrix of order $n$.

Every matrix can be represented as a linear combination of rank-one matrices. Thus, the attention matrix $\mathbf{M} \in \mathbb{R}^{l \times l}$ can be interpreted as an SSM with a state size of up to $l \gg n$. Consequently, we can employ prior research on time-varying model order reduction [9, 26, 38] to reduce $\mathbf{M}$ to an SSM with a smaller state size $n$. Specifically, we utilize the following SVD-based approximation:

---

**Algorithm 1** Approximation of Attention Matrix $\mathbf{M}$ as an SSM with State Size $n$

---

**Input:** Attention matrix $\mathbf{M} \in \mathbb{R}^{L \times L}$, state size $n$
**Output:** Approximated attention matrix $\tilde{\mathbf{M}} \in \mathbb{R}^{L \times L}$
**Procedure:**
1. For $k = 1, \ldots, L-1$:
    1.1 Define $H_k$ as the submatrix of $\mathbf{M}$ below and to the left of entry $M_{k,k}$:

$$H_k = \begin{bmatrix} M_{k,1} & \cdots & M_{k,k-1} \\ \vdots & \ddots & \vdots \\ M_{l,1} & \cdots & M_{l,k-1} \end{bmatrix}$$

    1.2 Perform the SVD on $\mathbf{H_k}$ and truncate it to rank $n$
    1.3 Integrate the truncated $\mathbf{H_k}$ back into the new matrix $\tilde{\mathbf{M}}$

---

Note that the diagonal elements of $\mathbf{M}$ were not subject to approximation in Algorithm 1 as they remain unchanged.

Although this approximation method for the semi-separable matrix is heuristic, it has been empirically shown to deliver good results. For further details on approximation methods for semi-separable matrices, and the theoretical background behind them, we refer the reader to [10, 26]

## C.2 Causal Low-rank Matrix Approximation

Given a set of self-attention matrices, we tried to find how close an causal low-rank matrix could approximate $M = \text{Softmax}(\mathbf{Q}\mathbf{K}^\top)$. To ensure the state size $N$, or in this case rank of $N$, of the approximation $\widetilde{M}$, we composed $\widetilde{\mathbf{M}} = \mathbf{L} \circ \mathbf{A}\mathbf{B}^\top$ where $\mathbf{A}, \mathbf{B} \in \mathbb{R}^{D,N}$, $\mathbf{L}$ is a $\mathbb{R}^{D,D}$ lower triangular mask, and $D = 512$.

We used the results from our causal low-rank (LR) experiments to inform much of our experimental design for later gradient descent-based approximations, which include both SSD classes (with and without $\mathbf{D}$ matrix) and RetNet. We experimented with various different low-rank approximation solvers. We found that gradient descent performed better than alternating gradient descent. Both types of gradient descent were better than alternating least-squares which often times reached less than optimal local minima. Causal low-rank matrix approximation can also be seen as a softer version of the low-rank matrix completion problem, but a semi-definite programming (SDP) approach was not able to outperform standard gradient descent.

Due to our LR approximation requiring gradient descent, we selected the number of steps in relation to the time required to calculate the semi-separable approximation of the same matrix. Given the heuristic approach for converting self-attention matrices to a semi-separable form (Appendix C.1) and its ability to be parallelized, we selected the number of steps for gradient descent based on the time it took to run an entire batch of matrices (32) using gradient decent on causal low-rank versus one matrix using the semi-separable heuristic. After testing with the state sizes $N = 16, 32, 64$, we found that 10,000 steps suitable as it was around a factor of $5\times$ compared to SSM. The 10,000 steps was maintained across all gradient based approximation classes (SSD, SSD without $\mathbf{D}$, and RetNet). Experiments using the finalized step count showed AdamW provided better results compared to SGD/Adam, and the use of a scheduler provided little gain.

During the experiments, we also found that initialization of the matrices $\mathbf{A}, \mathbf{B}$ played a significant role in the resulting approximation difference. The original $\mathbf{A}, \mathbf{B}$ values were sampled from $[0, 1)$; however, given $\forall M_{ij} \leq 1, i, j \in [D]$ due to the SoftMax operator, $\mathbf{A}, \mathbf{B}$ values was then sampled from $\left[0, \frac{1}{\sqrt{512N}}\right)$ to have the last row of the self-attention matrix be uniform probability. We then proceeded to vary the factor of the range exponentially, testing $\left[0, \frac{1}{\sqrt{512N}}\right) * 2^{\{-2,-1,0,1,2,4,8\}}$ where we found $\left[0, \frac{1}{\sqrt{512N}}\right) * 2^4$ provided the best initialization across multiple datasets. A normal distribution with $\mu = 0$ and $\sigma^2$ with the above tested values performed worse than the uniform distribution. Initialization experiments were conducted using the AdamW optimizer with a learning rate of 0.001 and the standard 10,000 steps. This and subsequent gradient descent classes use the same initialization for their $\mathbf{A}, \mathbf{B}$ matrices.

For all gradient descent experiments in Table 6, Three learning rates $0.1, 0.01, 0.001$ and AdamW were used for each combination of matrix class, state size, and dataset, with the best approximation being documented. The Frobenius matrix norm was used as the loss function.

---

**Listing 2** PyTorch example for generating Causal Low-rank approximation.

```
n_states = 512
state_size = 16 # or 32, 64
num_heads = 32

A = torch.rand((num_heads, n_states, state_size))
B = torch.rand((num_heads, state_size, n_states))

L = torch.tril(torch.ones((n_states, n_states)))

M_approximation = L * (A @ B)
```

---

## C.3 State Space Dual (SSD) Approximation

For the SSD approximation, we utilize the scalar SSM recurrent (1SS) representation introduced in [8]. A key component is the values of $a$, which we will refer to $l$ from here on out to avoid confusion with matrix $\mathbf{A}$, that constitute the final matrix mixer $\mathbf{M}$.

**Listing 3** PyTorch example for generating SSD (with and without $\mathbf{D}$ component) approximation.

```python
n_states = 512
state_size = 16 # or 32, 64
softplus = torch.nn.Softplus()
num_heads = 32

A = torch.rand((num_heads, n_states, state_size))
B = torch.rand((num_heads, state_size, n_states))
D = torch.rand((num_heads))

l = torch.rand((h, n_states))
L = torch.exp(segsum(softplus(l) * -1))

M_approximation = L * (A @ B)
if apply_D:
    M_approximation = M_approximation + torch.eye(n_states,n_states) * D[:, None, None]
```

Given the rolling multiplicative property of $L$ and the size of $\mathrm{n\_states}$, initialization of $l$ was important to prevent the bottom-right values of $L$ quickly reaching 0. We explored the uniform initialization of $[0, 1) + \{-10, -8, -6, -4, -2, 0, 2\}$ where smaller values of $l$ leads to less "decay" within the $L$ matrix. We found sampling $l$ from $[-8, -7)$ resulted in the best performance and use this initialization in the SSD family and RetNet class. As expected, adding the D component helps reduce the error between the approximation and actual attention matrix Table 6.

## C.4 RetNet Matrix Approximation

The Retention mechanism, introduced by [35], is a key component in RetNet models and can be represented mathematically as $(\mathbf{QK}^\top \cdot \mathbf{L})\mathbf{V}$. Here, the matrix $\mathbf{L}$ is defined element-wise by

$$
L_{nm} = \begin{cases} \gamma^{n-m}, & n \geq m \\ 0, & n < m \end{cases} \tag{7}
$$

where $\gamma$ is a decay factor. This lower triangular matrix $\mathbf{L}$ captures the temporal dependencies by decaying past values with respect to the current position.

In our approximation, we replace the product $\mathbf{QK}$ with matrices $\mathbf{A}$ and $\mathbf{B}$. The matrix $\mathbf{L}$ can be efficiently constructed in PyTorch using the following code, which generates a RetNet approximation:

**Listing 4** PyTorch example for generating the RetNet matrix approximation.

```python
n_states = 512
state_size = 16 # or 32, 64
softplus = torch.nn.Softplus()
num_heads = 32

A = torch.rand((num_heads, n_states, state_size))
B = torch.rand((num_heads, state_size, n_states))

l = torch.rand((num_heads))
L = torch.exp(segsum(softplus(einops.repeat(l, 'h -> h n', n=n_states) * -1)))

M_approximation = L * (A @ B)
```

This implementation provides a practical method for simulating the Retention mechanism, crucial for reducing computational complexity in RetNet models.

## C.5 Toeplitz Approximation

Our Toeplitz approximation technique calculates the matrix approximation by setting the value of each band of the Toeplitz matrix as the average of the values of the respective band in the attention matrix. Since each band in a Toeplitz matrix is constant along its diagonal, this method ensures that the approximation preserves the structure of the original matrix while maintaining computational efficiency.

To justify this approach, we observe that taking the mean per band minimizes the L2 norm (i.e., the sum of squared differences) between the original attention matrix and the approximated Toeplitz matrix. Specifically, for each band, the optimal value that minimizes the L2 difference between the two matrices is the average of the elements in that band. This is because the mean is the value that minimizes the sum of squared deviations for a set of numbers. As such, using the mean ensures that the approximation is as close as possible to the original matrix in terms of L2 distance, thereby providing a robust and efficient approximation method.

As before, we assume that the approximation is input-dependent, meaning that each attention matrix has its own unique Toeplitz approximation.

## C.6 Segsum Operator

The segsum operator computes the sum of elements across specified segments of a matrix, which, as applied in Appendices C.3 and C.4, corresponds to summing over the columns. This operation is crucial for various matrix manipulations, including the computation of the state-space dual (refer to Equation (2)). Below is the Python implementation of the 'segsum' operator using PyTorch.

---

**Listing 5** PyTorch implementation of the Segmented Summation (segsum) operator.

```python
def segsum(x):
    """Naive segment sum calculation. exp(segsum(A)) produces a 1-SS matrix,
        which is equivalent to a scalar SSM."""
    T = x.size(-1)
    x_cumsum = torch.cumsum(x, dim=-1)
    x_segsum = x_cumsum[..., :, None] - x_cumsum[..., None, :]
    mask = torch.tril(torch.ones(T, T, device=x.device, dtype=bool), diagonal=0)
    x_segsum = x_segsum.masked_fill(~mask, -torch.inf)
    return x_segsum
```

---

