# OpenReview forum: "Transformers to SSMs: Distilling Quadratic Knowledge to Subquadratic Models"
_NeurIPS.cc/2024/Conference — NeurIPS 2024 poster_

### Official Review · Reviewer_Kdfj · 2024-06-19

**Soundness:** 3
**Presentation:** 2
**Contribution:** 3
**Rating:** 5
**Confidence:** 4

**Summary:**

This paper proposes MOHAWK, a three-stage distillation method to transfer knowledge from pretrained transformer models to subquadratic models such as Mamba-2. The key idea is to view both transformers and SSMs as applying different forms of mixing matrices over token sequences. Experiments demonstrate that Phi-Mamba, a variant of Mamba-2 architecture tailored for distillation, achieves strong performance on downstream tasks using less than 1% of the training data compared to training from scratch.

**Strengths:**

1. The paper introduces an effective way to distill knowledge from pretrained transformer models to subquadratic models, enabling the linear attention community to obtain better models at a lower cost.
2. Mamba-2 aligns well with self-attention by converting the forget gate of GLA into a data-dependent value, achieving linear attention-like structures by simply removing the softmax and adding a forget gate.
3. The three-stage distillation process is well-designed and ablation studies demonstrate the importance of each stage, providing valuable insights for future research in this direction.

**Weaknesses:**

1. The paper does not provide a detailed comparison between the proposed distillation method and direct fine-tuning approaches like SUPRA (reference 21). While the three-stage distillation process achieves better performance with only 2.8B tokens, it is crucial to consider the additional computational overhead introduced by the need for both a teacher model and a student model during distillation. In contrast, direct fine-tuning methods, though requiring more tokens (e.g., +20B in SUPRA), may be simpler and more efficient in practice. The authors should discuss the trade-offs between these approaches and provide ablation studies to justify their choice of distillation over direct fine-tuning.

2. There are several typos throughout the paper. For example, in line 159, "Mamba-2 2" should have a "Figure" mentioned before the second "2". The authors should carefully proofread the manuscript to address these issues.

3. There is an inconsistency in the reported model size of Phi-1.5. The official documentation states that Phi-1.5 has 1.3B parameters, while this paper mentions a model size of 1.5B. The authors should clarify this discrepancy and ensure consistency throughout the paper.

4. The paper does not report the performance of the Phi-Mamba model on the MMLU benchmark, which is a standard evaluation for the Phi-1.5 model. Given that Phi-1.5 (1.3B) has reported MMLU performance, it would be informative to compare the MMLU scores of the Phi-Mamba model to assess its performance on a wider range of tasks.

**Questions:**

See weaknesses part.

**Limitations:**

The authors have discussed the potential limitations in Section 6.

---

> ### Author Rebuttal · Authors · 2024-08-07
>
> We appreciate the reviewer finding our method well-designed and the insights from the ablations valuable. We respond to the reviewer’s questions and concerns, which mainly focus on additional evaluations and comparisons to other distillation methods, below.
>
> > The paper does not provide a detailed comparison between the proposed distillation method and direct fine-tuning approaches like SUPRA (reference 21). While the three-stage distillation process achieves better performance with only 2.8B tokens, it is crucial to consider the additional computational overhead introduced by the need for both a teacher model and a student model during distillation. In contrast, direct fine-tuning methods, though requiring more tokens (e.g., +20B in SUPRA), may be simpler and more efficient in practice. The authors should discuss the trade-offs between these approaches and provide ablation studies to justify their choice of distillation over direct fine-tuning.
>
> Despite MOHAWK’s overhead, MOHAWK provides better performance than other cross-architectural distillation methods, as SUPRA’s minimum gap between SUPRA distilled models and their teacher models is greater than 6 percentage points. In addition, SUPRA is confined to converting attention into RNN-based models; however, MOHAWK can work with any architecture that can be represented using matrix mixers, as seen in our ablations in Table 4 when we distilled a Toeplitz and causal low-rank based Phi model. Compared to a more standard technique of weight transfer and knowledge distillation, we show in the second table in our shared response that only weight transfer and knowledge distillation leads to larger performance gaps in downstream metrics compared to the MOHAWK method.
>
> > There are several typos throughout the paper.
>
> > There is an inconsistency in the reported model size of Phi-1.5.
>
> We thank the reviewer for pointing out the inconsistency of the Phi-1.5 model. As mentioned in the shared notes, we have fixed it and the other typos in the paper.
>
> > The paper does not report the performance of the Phi-Mamba model on the MMLU benchmark, which is a standard evaluation for the Phi-1.5 model.
>
> The metrics we reported are a standard subset of the ones reported in the baseline papers in both Figure 1 and Table 1 of the paper (Mamba, xLSTM, RWKV, GLA, etc.). These seldom report MMLU, partly because it is known that it is hard to achieve above-random-guessing at the 1B model scale. Phi-1.5 has above-chance on MMLU which is hypothesized to be a byproduct of the special dataset that Phi-1.5 is trained on. We believe that disentangling the role of data is an important consideration for our distillation methods that we would like to explore in future work; however, we were unable to control this systematically at the 1B model scale due to a lack of strong and open-weight/open-data models.

---

> > ### Comment · Reviewer_Kdfj · 2024-08-12
> > **Thank you**
> >
> > The reviewer thanks the authors for the discussion. It solved most of my concerns. I decide to keep my score.

---

### Official Review · Reviewer_2yXa · 2024-07-10

**Soundness:** 3
**Presentation:** 1
**Contribution:** 2
**Rating:** 6
**Confidence:** 4

**Summary:**

The paper distills a Transformer model into the Mamba architecture by using about 1% of the pertaining dataset.

**Strengths:**

1. The paper successfully distills a Transformer model into the Mamba architecture by using about 1% of the pertaining dataset, reaching best performance on some of the tasks when compared with other SSM models and is close to the performance of the teacher model for most tasks.
2. The authors compare against a large number of SSM models.
3. The authors have performed extensive ablations to ensure that all three steps they propose are necessary to achieve good distillation.

**Weaknesses:**

1. The paper uses the term "mixing matrices" repeatedly throughout the paper. However, this term is never formally defined, except as the matrix M in Section 3.1, although the next paragraph explains that such matrix multiplication is actually not used in practice.
2. The description of the "Matrix Orientation" step in Section 4.1 is also not at all clear. What does "matrix mixer layer" mean? What are TeacherMixer and StudentMixer? Is this done layer by layer, block by block or are they all aligned at the same time? What's the $x$ that we are minimizing over? Is it full sequences (previously denoted with capital $X$)? What are the dimensions of the models?
3. Similar issues are present in the description of the "Hidden-State Alignment" step in Section 4.2. There is also a new term $\mathbf u$ that appears but is never defined.
4. No code or trained model artifacts were provided nor were mentioned that will be provided. The authors declare in the Checklist that "Our proposed method is quite simple and all significant details are included", however, due to the above points, I do not think that replicating their experiments would be at all straightforward.
5. Minor comment: On line 108, matrix multiplication has complexity larger than $O(T^2)$.
6. Minor comment: On line 48, "strong" -> "stronger"



Overall, while the experimental results seem impressive, the rest of the paper is not clear and do not allow for the reproducibility of the results. I would recommend that the authors clearly and formally describe the architectures of the teacher and student networks and explicitly define the objective functions that they optimize for each of the three stages.

**Questions:**

See Weaknesses.

**Limitations:**

The paper offers only one sentence of limitations in the Discussion section and it is mostly about the need for further experiments. Missing limitations are: difficulty to reproduce, investigation of only one pair of teacher and student, and failing to achieve the performance of the teacher model.

There is no potential negative societal impact to be addressed.

---

> ### Author Rebuttal · Authors · 2024-08-07
>
> We are glad the reviewer found our experimental results impressive and liked our extensive comparison to other SSM and sub-quadratic models. The reviewer’s main concern is the presentation, clarity, and terminology. Based on our shared response, we have fixed the issues with the presentation, and the terminology used in this paper is standard and based on an existing line of work [1-3]. To make the paper self-contained, we decided to summarize some of the key terms in our Preliminaries section. We appreciate the reviewer for their comments which have led us to make the paper more readable and self-contained, and we will try to clarify the questions left further below.
>
> >  The paper uses the term "mixing matrices" repeatedly throughout the paper. However, this term is never formally defined, except as the matrix M in Section 3.1, although the next paragraph explains that such matrix multiplication is actually not used in practice.
>
> A mixing matrix/matrix mixer is any matrix that represents a sequence transformation when applied to an input sequence X. Hence, the definition of a mixing matrix is quite broad, as many algorithms, such as self-attention, Mamba, and other structured matrices, can be viewed as applying their unique matrix mixer to the input. In general, one can use naive matrix multiplication to transform an input sequence with a matrix mixer; however, certain classes of matrix mixers, like Toeplitz or Mamba, have special structures that allow for more efficient matrix multiplication.
>
> > What does "matrix mixer layer" mean?
>
> The matrix mixer layer is the layer that “hosts” the matrix mixer. In the case of the Transformer Phi teacher model, this would refer to the self-attention layer, which includes the input projections, the actual self-attention mechanism, and the output projection. For the Phi-Mamba student model, this would refer to the Mamba-2 layer, which encompasses the projections, convolution, SSM mechanism, gate, output projection, etc.
>
> > What are TeacherMixer and StudentMixer? Is this done layer by layer, block by block or are they all aligned at the same time?
>
> We refer to the TeacherMixer and StudentMixer as the extracted matrix mixer from the matrix mixer layer. The Matrix Orientation step has the student model learn to approximate the teacher’s mixing matrix; in Phi-Mamba’s case, the stage would be minimizing the difference between the “unraveled” SSM matrix mixer, i.e., StudentMixer, (Equation 2 in the paper) and the self-attention matrix Softmax(QK.T), i.e., TeacherMixer, found in the respective teacher layer.
>
> This is done layer-by-layer but can be seen as block-by-block in our Phi case, as each Phi block only has one self-attention matrix mixer layer.
>
> > What's the $x$ that we are minimizing over? Is it full sequences (previously denoted with capital $X$)? What are the dimensions of the models?
>
> $\textbf{x}$ is the input to the teacher matrix mixer layer which is then set as the input to the respective student layer as well. The input is passed through both layers and then the difference between the extracted matrix mixer is minimized. The layer parameters for the student layer, $\phi$, are then updated via gradient descent. The dimensions of the model match Phi-1.5 in terms of layers, state size, attention heads, etc.
>
> > Similar issues are present in the description of the "Hidden-State Alignment" step in Section 4.2. There is also a new term $\mathbf{u}$ that appears but is never defined.
>
> The Hidden-State Alignment aims to minimize the difference between the output of the entire block, not just the matrix mixers. In our case, $\mathbf{u}$ is the same as $\mathbf{x}$, but depending on the design of the teacher block, this might not always be the case, so we decided to utilize a different variable. We apologize for any confusion this may have caused.
>
> > No code or trained model artifacts were provided nor were mentioned that will be provided.
>
> As mentioned in the shared response, we are planning to publicly release the model code and the pretrained weights. The code for our model architecture will contain certain flags that allow for the return of elements, such as the matrix mixer of Mamba-2, which will make replicating the experiments easier.
>
> > Minor comment…
>
> As mentioned in the shared response, we have fixed the presentation issues, e.g., typos, broken links, and unclear portions, of the paper.
>
> [1] https://arxiv.org/abs/2405.21060
>
> [2] https://arxiv.org/abs/2407.09941
>
> [3] https://arxiv.org/abs/2402.18668

---

> > ### Comment · Reviewer_2yXa · 2024-08-12
> >
> > Thank you for the response! In light of most of my concerns were related to the clarity of the presentation and the authors saying that they would improve that, I increased my score. This is an interesting paper with very good results and significant potential for impact. However, whether this is realized or not depends on how easy it would be for others to apply your methods. For this reason, I would strongly urge the authors to take the opportunity to improve the presentation for the camera-ready version.

---

### Official Review · Reviewer_1qnq · 2024-07-10

**Soundness:** 3
**Presentation:** 2
**Contribution:** 3
**Rating:** 6
**Confidence:** 4

**Summary:**

The authors consider the problem of distilling transformers into SSM models (Mamba in particular), which results in the reduction of quadratic complexity at inference to subquadratic complexity. For this purpose, the authors propose MOHAWK, a method consisting of several steps, each aiming to match a different aspect of SSMs and transformers. The method is then tested on the Phi-1.5 transformer model and distilled into Phi-Mamba proposed by the authors. The paper shows that the resulting model performs very well and presents a thorough analysis that each step of their method is important.

**Strengths:**

- The paper considers a novel and important problem of distilling knowledge from quadratic models to subquadratic models. This could reduce the inference costs and, as such, is of significant practical importance
- The empirical results obtained by the proposed method are quite good. The authors also provide a detailed analysis of the impact of each step. The authors quite convincingly show that fine-grained matching of particular blocks is needed before we start end-to-end knowledge distillation.
- To the best of my knowledge, the paper faithfully discusses the related work.

**Weaknesses:**

- I have some reservations about the empirical evaluation:
    - The crucial consideration in the large language model domain is the scaling properties. Given the current experiments, I’m not convinced that the proposed method will scale well with data and available compute. It’s obviously nice that the approach works well even when using 1% of the data from other models, but how much data would we need to close the gap to Phi-1.5 completely, or up to negligible levels? What are the fundamental limitations of this approach, what Mamba cannot do that transformers can?
    - Also, will it work with models larger than 1.5B? For example, can we use this approach to scale Mamba up beyond its standard regime (e.g., 7B, 70B)?
    - The Phi architecture used in this paper works well, but I would be curious to see how MOHAWK performs with more standard transformer architectures (e.g., LLAMA).
    - There’s a confounding factor - Phi is trained on a very well-curated dataset, which makes it perform very well for its size. The Pile dataset used for training Mamba and other models is not that good. For a fair comparison, I think one should use models trained on (roughly) the same data as baselines.
- There are certain problems with the presentation in this paper:
    - There are numerous typos and grammar errors, see below.
    - The way the Tables and Figures are referenced in the text is confusing. Some of the links are broken, and others are not referenced when they should be. In particular, Section 5 is difficult to understand because of that.

Minor issues/typos:
- Line 27: “raises a natural question: it is” → is it?
- Line 33: “differen”
- Line 48: “benchmarks strong than” → stronger than
- Line 116: “$A_t h_t$ is the identity matrix I multiplied by a scalar αt” -> $A_t$, I don't think you need the $h_t$ term there?
- Figure 3: “Despiting training” → Despite training
- And many more I think. I suggest carefully proofreading the manuscript.

**Questions:**

Please see the weaknesses section above. I'm particularly interested in the scaling properties of MOHAWK, and its fundamental limitations. How far can we go?

**Limitations:**

The limitations are not clearly discussed. Issues that should be mentioned include:
* How well the method will scale?
* How well does it work with more standard transformer/mamba architectures (not Phi)?
* What is the impact of the data?

---

> ### Author Rebuttal · Authors · 2024-08-07
>
> We appreciate the reviewer finding our analysis thorough and our method novel and important. The reviewer also raised an important point about the entanglement of the architecture, data, and MOHAWK method when looking at the distillation results. We would like to preface by reiterating that our key contribution in this paper is the MOHAWK method which enables the effective distillation from Transformers to alternative architectures, and distilling from Phi-1.5 to a Mamba model is validation of the method’s effectiveness. We will structure the following response by disentangling the architecture first and then the data from the method.
>
> > What are the fundamental limitations of this approach, what Mamba cannot do that transformers can?
>
> **The limitations of the Mamba architecture is an orthogonal line of work** [1-5]. Our distillation method is not directly tied to Mamba, since we have distilled to other architectures in Table 4, and as stated in the shared response, our key contribution is the distillation method while Phi-Mamba is our validation of the effectiveness of the method. Our paper’s goal is to introduce a distillation method that can create performant alternative architecture-based models which we show via the strong downstream metrics of Phi-Mamba and the ability of MOHAWK to fully distill Phi-from-Phi.
>
> > how MOHAWK performs with more standard transformer architectures (e.g., LLAMA)
>
> **MOHAWK should work with any architecture that has a valid matrix mixer representation**. With more standard Transformer architecture like Llama, the matrix mixer layer is similar to that of Phi’s, which means that Stage 1 and Stage 2’s strong performance should translate. The reason we did not use a more standard Transformer architecture was that there are few strong and open-weight models at the 1B scale.
>
> > Also, will it work with models larger than 1.5B? For example, can we use this approach to scale Mamba up beyond its standard regime (e.g., 7B, 70B)?
>
> We believe that our MOHAWK method will be able to scale to larger models but will require scaling to industrial resources. We distilled to a 1.5B model to validate our new distillation method on a smaller scale. The next step would be to distill a 7-8B model, but that is beyond the scope of this work. However, **preliminary evidence** (in Section 5.4) on just Stage 1 **indicates potential for scaling to larger models** as the experiments show that the SSD matrix mixer can effectively approximate the attention matrices of a Llama-7B model.
>
> > how much data would we need to close the gap to Phi-1.5 completely, or up to negligible levels?
>
> In the second table of the shared response, we can **close the gap to Phi-1.5 completely with only 5 billion tokens** if the student model is also a Transformer architecture. We believe that the ability to close the gap with the teacher model stems from the differences in expressiveness between the teacher and student matrix mixers and not from the MOHAWK distillation method itself.
>
> > There’s a confounding factor - Phi is trained on a very well-curated dataset, which makes it perform very well for its size. The Pile dataset used for training Mamba and other models is not that good. For a fair comparison, I think one should use models trained on (roughly) the same data as baselines.
>
> When comparing our Phi-Mamba to the Transformer-Mamba hybrid Samba-1.7B and Mamba-SWA-MLP 1.6B model, which are trained on a stronger dataset (the Phi-2 dataset) compared to the student model and a comparable dataset compared to the teacher model, we achieve comparable performance with less than 2% of their reported 230B total training tokens without employing any Transformer based blocks. The Samba model uses the same model dimension but uses 12 attention-based layers while the Mamba-SWA-MLP model uses 18 attention-based layers. In addition, our Phi-Mamba is comparable to a Mamba-1.8B trained with the aforementioned Phi-2 dataset [6]. When compared to the other alternative models that use The Pile or SlimPajama, we still outperform them when the datasets are reported to be somewhat similar compared to C4 (all three are within ~2 percentage points) [7].
>
> |Model|WinoG.|ARC-E|ARC-C|PIQA|HellaS.|Avg.↑|
> |-----|------|-----|-----|----|-------|-----|
> |Phi-1.5 1.3B|73.4|75.6|48.0|76.6|62.6|67.2|
> |Phi-Mamba 1.5B|71.7|74.0|44.1|75.5|60.2|65.1|
> |Mamba (Phi2 Dataset) 1.8B|73.4|78.0|45.2|77.3|49.8|64.7|
> |Mamba-SWA-MLP 1.6B|73.7|76.7|46.2|76.5|49.7|64.6|
> |Samba 1.7B|72.9|79.3|48.2|77.1|49.7|65.4|
>
> > There are certain problems with the presentation in this paper:
>
> > Minor issues/typos:
>
> As mentioned in the shared response, we have addressed these.
>
> [1] https://arxiv.org/abs/2404.08819
>
> [2] https://arxiv.org/abs/2402.04248
>
> [3] https://arxiv.org/abs/2402.03170
>
> [4] https://arxiv.org/abs/2402.01032
>
> [5] https://arxiv.org/abs/2402.18510
>
> [6] https://arxiv.org/pdf/2406.07522
>
> [7] https://arxiv.org/abs/2406.17557

---

> > ### Comment · Reviewer_1qnq · 2024-08-12
> >
> > Thank you for the thorough response. After reading the rebuttal and other reviews, I decided to change my score to 6 (Weak Accept).

---

### Author Rebuttal · Authors · 2024-08-07

We thank all reviewers for their time spent reading our paper and providing thoughtful comments and feedback. All reviewers agree that our method enables cost-effective cross-architectural distillation, which is validated by our strong downstream performance, and that the careful ablations support our claim that all three stages of MOHAWK are crucial for effective distillation. The main weaknesses that have been brought up mainly surround the reproducibility and presentation of the paper. We have used these comments, questions, and concerns to improve our submission. In addition, we directly answer them in both the shared and individual responses and provide new experiments and analyses we would like to share:

- Improving results and reproducibility: We describe our new training regime to **improve reproducibility** and distill a **new version of Phi-Mamba 1.5B using only 3B tokens**, achieving an **average metric score of 62.6**, more than 1.2 percentage points than in the original submission.
- Disentangling MOHAWK from Architecture: We ran additional baselines where we **distill from a pre-trained Phi-1.5 to a new Phi-1.5** to show the effectiveness of the MOHAWK distillation method compared to other potential alternatives and **highlight the role of the distillation method vs that of the architecture**.
- Paper presentation and model release: We have addressed all presentation issues raised by reviewers and plan to **open-source all model code and final model weights**.

To recap, MOHAWK is a method that can distill a pre-trained Transformer mode into any alternative architectures that can be represented using matrix mixers. The three stages progressively distill the Transformer architecture by first matching the mixing matrices, then the hidden states at the end of each mixer layer, and finally the end-to-end logits. MOHAWK is validated by our Phi-Mamba 1.5B model that demonstrates substantially stronger performance than all past open-source non-Transformer models at similar sizes while using less than 1% of the typical pre-training token count.
## Improving Results and Reproducibility
Reviewer 2yXa has mentioned increasing the reproducibility of our distillation process. Below is a paraphrased excerpt from our appendix in the revised paper.

> To train the final model, we use the AdamW optimizer with $\beta = (0.9, 0.95)$, a weight decay of 0.1 with gradient clipping of 1.0, and a Warmup-Stable-Decay (WSD) scheduler featuring 10\% warmup and 10\% decay with a linear warmup and linear cooldown function. Automatic mixed precision training to bf16 was used for all three stages as well. Stage 1 used $\text{batch size}=2^{15}, \text{lr}=5\times 10^{-4}$, Stage 2 used $\text{bs}=2^{15}, \text{lr}=2\times 10^{-3}$, and Stage 3 used $\text{bs}=2^{19} \approx 0.5\text{M}, \text{lr}=5\times 10^{-4}$. These values were determined via sweeping over the hyperparameters, where the exact process is explained in the new appendix. The previous model was trained with a constant learning rate of $1 \times 10^{-4}$ for all three stages with a functional batch size of $2^{16}$ for Stages 1 and 2 and $2^{18}$ for Stage 3. All other components were the same.

Based on the revised training regime detailed above, we trained a refined version of Phi-Mamba 1.5B using only 300 million more tokens (3.0B tokens total) by distributing the 3B C4 tokens into 80M/160M/2.76B splits for Stage 1/2/3. The new model is **strictly better on all metrics than our previously reported Phi-Mamba**.
| Model|Tokens / Dataset|WinoG.|Arc-E|Arc-C|PIQA|HellaS.|Lamb.|Avg. ↑|
|-|-|-|-|-|-|-|-|-|
|Phi-1.5-1.3B|150B / unknown|73.4|75.6|48.0|76.6|62.6|53.4|64.9|
|**New Phi-Mamba-1.5B**|3.0B / C4|**71.7**|**74.0**|**44.1**|**75.5**|**60.2**|50.1|**62.6**|
|Old Phi-Mamba-1.5B|2.7B / C4|69.1|73.5|43.8|74.7|59.3|48.2|61.4|
|Mamba-2-1.3B|315B / The Pile|60.9|64.3|33.3|73.2|59.9|**65.7**|59.6|
## Disentangling MOHAWK from Architecture
MOHAWK is a high-level method for cross-architectural distillation. Reviewers have noted that the effectiveness of the method has been conflated with the impact of the student model’s architecture. To address this, we conducted a baseline experiment where the student model architecture is fixed to match Phi-1.5. This experiment demonstrates the **effectiveness of MOHAWK when distilling from Phi to Phi** using the same budget of 5B tokens and the same hyperparameters as the previously trained model, as detailed in the section above.

|Stages Performed|WinoG.|ARC-E|ARC-C|PIQA|HellaS.|Lamb.|Avg.|
|-|-|-|-|-|-|-|-|
|2|71.0|77.1|40.8|77.8|60.9|51.3|63.1|
|3|64.1|73.7|38.7|58.6|75.6|48.0|59.9|
|2-3|69.8|77.4|44.8|78.2|61.3|54.4|64.3|
|1-3|74.2|76.7|43.5|78.6|61.7|54.2|64.9|

Using solely Knowledge Distillation and Weight Transfer (Stage 3) can only recover parts of the original Phi’s performance (59.9 avg) while adding Stage 2 can improve it significantly (64.3). **Only with all stages of MOHAWK is the overall performance restored**, highlighting the effectiveness of our MOHAWK method over traditional knowledge distillation alternatives, and showing that **performance gaps in Phi-Mamba are due to architectural differences between Mamba and Transformer**. Understanding the fundamental limitations of subquadratic vs quadratic models is an active area of research that is independent of our distillation method, where we use MOHAWK to match or exceed the best trained-from-scratch sub-quadratic models.
## Paper Presentation and Model Release
All reviewers highlighted issues with the presentation, which we have since addressed for the camera-ready version, including typos, grammatical errors, and broken links. We thank reviewer Kdfj for noting that the base Phi-1.5 model is 1.3B, while the new Phi-Mamba model is 1.5B. Additionally, **we plan to open-source all model code and the final Phi-Mamba 1.5B model weights**.

---

### Decision · Program_Chairs · 2024-09-25

**Decision:**

Accept (poster)

**Comment:**

The work provides a novel method for distillation from Transformer based models to Mamba based SSMs. The method performs progressive distillation of the mixing mechanism of Transformers into an SSM, by applying a multi-stage algorithm for converting the token mixing of Transformers, which operates in quadratic time, into a sub-quadratic model. The authors demonstrate the efficacy of their method by distilling phi-1.5, a small Transformer, into an equivalent Mamba-based model, with very small amount of training data.

The reviewers raised some concerns regrading the empirical evaluation in the paper, the presentation of the results and pointed some methodological problems. However, it seems that the authors addressed most of the concerns in the rebuttal period. In light of this, I recommend that this paper is accepted.